# Exploiting Negative Samples: A Catalyst for Cohort Discovery in Healthcare Analytics

## Abstract

Healthcare analytics, particularly binary diagnosis or prognosis problems, present unique challenges due to the inherent asymmetry between positive and negative samples. While positive samples, representing patients who develop a disease, are defined through rigorous medical criteria, negative samples are defined in an open-ended manner, resulting in a vast potential set. Despite this fundamental asymmetry, previous research has underexplored the role of negative samples, possibly due to the enormous challenge of investigating an infinitely large negative sample space. To bridge this gap, we propose an approach to facilitate cohort discovery within negative samples, which could yield valuable insights into the studied disease, as well as its comorbidity and complications. We measure each sample's contribution using data Shapley values and construct the Negative Sample Shapley Field to model the distribution of all negative samples. Then we transform this field via manifold learning, preserving the data structure information while imposing an isotropy constraint in data Shapley values. Within this transformed space, we identify cohorts of medical interest through density-based clustering. We empirically evaluate the effectiveness of our approach on our hospital's electronic medical records. The medical insights revealed in the discovered cohorts are validated by clinicians, which affirms the medical value of our proposal in unveiling meaningful insights consistent with existing domain knowledge, thereby bolstering medical research and well-informed clinical decision-making.

## 1 Introduction

Healthcare analytics leverages diverse healthcare data sources to perform many analytic tasks including diagnosis [28] and prognosis [35]. Electronic Medical Records (EMR) are perhaps the most important of these data sources, since they play a crucial role in recording patients' essential information and providing a comprehensive view of their health conditions. The recently increasing availability of EMR data has spawned the development of healthcare analytics models for effective patient management and medical resource allocation.

Without loss of generality, let us delve into a diagnosis or prognosis problem of predicting whether a patient has developed/will develop a certain disease based on the EMR data. This problem is a binary classification, where patients who develop the disease are "positive samples", while those who do not are "negative samples". Notably, we identify the unique nature of such binary classifications in healthcare analytics, as compared to traditional classification tasks. For instance, when classifying cats vs. dogs, both positive and negative samples are based on objective facts. However, in healthcare analytics, positive samples are defined according to rigorous medical criteria, based on medical theories and experience. Contrarily, negative samples are defined in an unrestricted manner, as the complementary set of the positive samples. Consequently, the set of negative samples may encompass a vast number of diverse individuals who are outside the scope of the studied disease or who are healthy. This leads to an inherent asymmetry between positive and negative samples, as positive samples are well-defined and bounded, while negative samples are diverse and open-ended.

Submitted to 37th Conference on Neural Information Processing Systems (NeurIPS 2023). Do not distribute.

Despite such fundamental asymmetry in healthcare analytics, previous research has not adequately addressed the role of negative samples. One potential reason for this research gap is the enormous challenge posed by investigating an infinitely large negative sample space, which cannot be easily addressed using existing approaches, e.g., it could be difficult to understand why general healthy individuals do not develop a disease. Nonetheless, it is crucial to probe into negative samples for a more comprehensive investigation of the studied disease. Although it may not have developed in these samples, some may exhibit similar symptoms or even develop related conditions such as its comorbidity or complications. Hence, these negative samples are in urgent need of close medical attention, as they provide an opportunity for clinicians to gain a deeper understanding of the studied disease, leading to more accurate and comprehensive diagnoses, prognoses, and treatment plans.

In this paper, we aim to address the gap by exploring negative samples in healthcare analytics. Given the diversity of negative samples, it may not be meaningful to consider them all as one "group". Instead, we examine the underlying distribution of negative samples to automatically identify medically insightful groups of patients with shared characteristics, referred to as "cohorts" [32, 49]. Such cohort discovery among negative samples can provide fresh insights to clinicians on the studied disease, e.g., comprehending the factors contributing to the absence of the disease and the development of related conditions.

As front-line clinicians and medical researchers, we bring a unique perspective to guide our methodology design in effectively discovering cohorts among negative samples. In Sec. 3, we elaborate on our approach with three components. Firstly, we propose to quantify each sample's contribution to the prediction task using data Shapley values [38, 12]. We then construct the Negative Sample Shapley Field, an inherently existing scalar field describing the distribution and characteristics of all negative samples (Sec. 3.1). Secondly, to effectively discover cohorts, we transform the original field by manifold learning [3] while preserving the original data structure information and ensuring that changes in data Shapley values are isotropic in all orientations (Sec. 3.2). Thirdly, in the transformed manifold space, we identify densely-connected clusters among the negative samples with high data Shapley values through DBSCAN (Sec. 3.3). These clusters help us locate "hot zones", which are our desired cohorts to discover, exhibiting similar medical characteristics with high data Shapley values.

**Our contributions are summarized below:** (i) We bridge the research gap caused by the asymmetry between positive and negative samples in healthcare analytics by exploring negative samples for cohort discovery. (ii) We propose an innovative approach for effective cohort discovery: constructing the Negative Sample Shapley Field, transforming the field by manifold learning with structure preservation and isotropy constraint, and discovering cohorts in the manifold space via DBSCAN. (iii) We empirically evaluate the effectiveness of our approach using our hospital's EMR (Sec. 4). The experimental results validate the efficacy of each component and demonstrate the capability of our approach for cohort discovery, unveiling meaningful insights that align with existing domain knowledge and have been verified by clinicians. These findings have the potential to benefit medical practitioners by facilitating medical research and clinical decision-making in healthcare delivery.

## 2   Problem and Our Solution

**Distinctiveness of negative samples and the unbounded negative sample space.** Let us take hospital-acquired acute kidney injury (AKI), a disease we strive to handle in practice, as an example. AKI is defined according to KDIGO criteria [19] based on a lab test, serum creatinine (sCr). The disease definition has two criteria: absolute AKI and relative AKI. Absolute AKI criterion is met when sCr exhibits a rise exceeding 26.5 umol/L within the last two days, whereas relative AKI is defined by a rise of sCr 1.5 times or higher over the lowest sCr value within 7 days. In this AKI prediction task, we aim to predict if a patient will develop AKI in the near future. A positive sample is a patient who meets the stringent criteria above, and hence, has a closed definition, whereas a negative sample has an open definition without restrictions. Hence, negative samples in nature form an unbounded space, demonstrating an asymmetry compared to positive samples.

**Construction of the Negative Sample Shapley Field for cohort discovery.** To facilitate the analysis of negative samples, we need to investigate their distribution and identify those that are most relevant to the prediction task (e.g., AKI prediction task above) and hence worth exploring. In this regard, we propose to measure the valuation of each negative sample to the task by its data Shapley value. Based on such valuations, we construct a scalar field, the Negative Sample Shapley Field, in which each point is a negative sample, and the point's value is its data Shapley value. This field depicts the

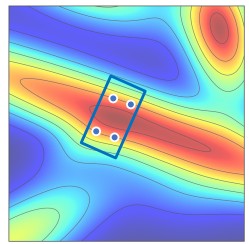 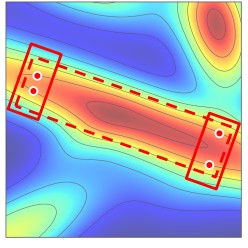 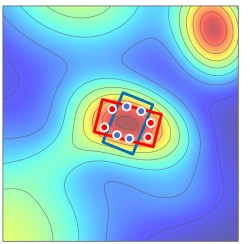

(a) Discovered hot zone in the Negative Sample Shapley Field by clustering high-value negative samples

(b) Mis-discovered hot zones in the Negative Sample Shapley Field

(c) Manifold space integrating data structure information and isotropy constraint

Figure 1: Discovery of hot zones in the Negative Sample Shapley Field.

distribution and characteristics of negative samples (see Figure 1(a) for an example). We define **"hot zones"** in this field, identified by points with high data Shapley values, as **"cohorts"**. Our objective is to automatically detect these cohorts, revealing medically meaningful patterns.

**Cohort discovery via manifold learning and density-based clustering.** We note that the vast number of negative samples renders an exhaustive search infeasible. Although the Negative Sample Shapley Field is continuously differentiable, the high computational overhead makes it intractable to find local optima via gradient descent. To overcome this obstacle, we make the assumption that a subset of negative samples collected in clinical practice carries significant medical value, e.g., patients who visit hospitals for examinations but do not develop the disease. We posit that these real-world negative samples should be proximate to our desired hot zones in the space and can effectively sample our hot zone boundaries, which are hence of medical interest.

In Figure 1, we exemplify how to discover hot zones in the Negative Sample Shapley Field. Figure 1(a) and (b) demonstrate four points situated on the same contour line, indicating their inclusion in the same hot zone. However, only the former case yields the expected discovered cohort, while the latter leads to mis-discovery. This highlights that the originally constructed Negative Sample Shapley Field is suboptimal for cohort discovery among negative samples, due to its anisotropy in data Shapley values. To overcome this issue, we propose a manifold learning approach. Specifically, we leverage manifold learning to reduce the dimensionality of the raw sparse EMR data to derive compact representations that not only preserve the underlying data structure information but also benefit subsequent spatial clustering analysis. Further, we introduce an isotropy constraint to ensure uniform changes in data Shapley values across all orientations, which prevents the mis-discovery as in Figure 1(b). This transformed space, integrating data structure information and the isotropy constraint, is more suitable for subsequent cohort discovery as illustrated in Figure 1(c).

Our objective is then to identify medically meaningful cohorts, specifically dense regions formed by negative samples with high data Shapley values in the manifold space. We set a data Shapley value threshold to extract negative samples with high values and employ the DBSCAN algorithm to detect the hot zones among them. The derived cohorts could shed light on the studied disease, its related comorbidity, and complications, thereby empowering clinicians in practical healthcare delivery.

## 3 Methodology

### 3.1 Negative Sample Shapley Field Construction

Given EMR data $\mathcal{D} = \{d_i\}$, where $d_i$ is a sample with $i \in \{0, \ldots, N-1\}$ and $N$ denotes the total sample number. We focus on binary classification, and each $d_i$ consists of input features and a binary label. To investigate negative samples for cohort discovery, we divide $\mathcal{D}$ into $\mathcal{D}^+$ and $\mathcal{D}^-$, representing positive and negative samples. We denote $\mathcal{D}^- = \{d_i^-\}$, where $d_i^-$ is a negative sample with $i \in \{0, \ldots, N^- - 1\}$ and $N^-$ is the negative sample number.

Each negative sample $d_i^- = (\mathbf{x}_i, y_i)$ comprises the input features $\mathbf{x}_i$ and its corresponding binary label $y_i$. Our objective is to measure the value of each negative sample by quantifying its contribution to the prediction performance, which we refer to as data valuation. Data Shapley value [12], stemming from Shapley value in cooperative game theory, has made significant advances in data valuation [38], which inspires our proposal to calculate the data Shapley value of each negative sample as its value. Specifically, let $F$ denote the prediction model and suppose we are interested in evaluating $F$'s

performance on a subset of negative samples $\mathcal{Q} \subseteq \mathcal{D}^-$, along with all the positive samples $\mathcal{D}^+$. We define $M$ as the performance metric function, and then $M(\mathcal{D}^+ \cup \mathcal{Q}, F)$ is the performance achieved on the combined set of $\mathcal{D}^+$ and $\mathcal{Q}$. We define $s_i$ as the data Shapley value for the negative sample $d_i^-$. $s_i$ satisfies three properties of Shapley values: (i) null player, (ii) symmetry, and (iii) linearity, which are the essential properties of an equitable data valuation [12]. We calculate $s_i$ as follows.

**Proposition 1** *The data Shapley value $s_i$ for a negative sample $d_i^-$ is given by:*

$$s_i = H \sum_{\mathcal{Q} \subseteq \mathcal{D}^- - \{d_i^-\}} \frac{M(\mathcal{D}^+ \cup \mathcal{Q} \cup \{d_i^-\}, F) - M(\mathcal{D}^+ \cup \mathcal{Q}, F)}{\binom{N^- - 1}{|\mathcal{Q}|}} \tag{1}$$

*where $H$ is a constant and the summation is taken over all subsets of negative samples, except $d_i^-$.*

As the computation of data Shapley value for negative samples has exponential complexity, we further employ Monte Carlo permutation sampling for approximation [6]. Let $\Pi$ represent a uniform distribution of all the permutations among $\mathcal{D}^-$, $s_i$ can be approximated as the following expectation:

$$s_i = E_{\pi \sim \Pi}[M(\mathcal{D}^+ \cup A_\pi^{d_i^-} \cup \{d_i^-\}, F) - M(\mathcal{D}^+ \cup A_\pi^{d_i^-}, F)] \tag{2}$$

where $A_\pi^{d_i^-}$ denotes all the negative samples before $d_i^-$ in a permutation $\pi$. By repeating this approximation, we can derive the estimated data Shapley value $s_i$ efficiently. After computing the data Shapley value of each negative sample, we define the Negative Sample Shapley Field below.

**Definition 1** *(Negative Sample Shapley Field) We define the Negative Sample Shapley Field $\mathcal{S}$ as an inherently existing scalar field representing the distribution of data Shapley values across all negative samples in space. In this field, each point denotes a negative sample $d_i^-$ and is associated with its data Shapley value $s_i$. Therefore, $\mathcal{S}$ is a mathematical function that maps the input of each negative sample to its corresponding data Shapley value: $\mathbf{x}_i \mapsto s_i$.*

With this field $\mathcal{S}$ constructed, our goal of cohort discovery within negative samples can be reframed as the task of identifying "hot zones" - grouped regions within $\mathcal{S}$ exhibiting high data Shapley values.

## 3.2 Manifold Learning with Structure Preservation and Isotropy Constraint

As in Figure 1(a) and (b), although we hope to detect a similarly clustered cohort in the Negative Sample Shapley Field in both scenarios, the anisotropic nature of the space, i.e., the non-uniform distribution of negative samples with similar data Shapley values, present significant challenges. To mitigate these challenges, we propose to employ manifold learning [3] to transform the original space $\mathcal{S}$ into a new geometric space $\mathcal{S}'$. As elaborated in Sec. 2, to avoid mis-discovery such as Figure 1(b), we should simultaneously preserve the underlying structural information in the data while imposing an isotropy constraint on the data Shapley values in $\mathcal{S}'$. The resulting $\mathcal{S}'$ will be more amenable to effective cohort discovery, enabling us to identify medically relevant cohorts more accurately.

We employ a stacked denoising autoencoder (SDAE) [44] as the backbone model for manifold learning and integrate the isotropy constraint while preserving the data structure information in $\mathbf{x}_i$. Autoencoders (AE) [23, 22] are well-known for capturing data structures by reconstructing input data. Denoising autoencoders (DAE) [43] are further developed to enhance the learned representations with the capability of handling input data corruption. By stacking multiple layers of DAE, SDAE can abstract higher-level robust representations. The model architecture is illustrated in Figure 2.

Consider an SDAE consisting of $K$ DAEs. For the $k$-th DAE ($k \in \{0, \ldots, K-1\}$), the encoder takes $\mathbf{h}_i^{(k)}$ as input, where $\mathbf{h}_i^{(0)} = \mathbf{x}_i$ corresponds to the original input. We define $\tilde{\mathbf{h}}_i^{(k)}$ as the corrupted version of $\mathbf{h}_i^{(k)}$ with masking noise generated by a stochastic mapping, $\tilde{\mathbf{h}}_i^{(k)} \sim g_\mathcal{D}(\tilde{\mathbf{h}}_i^{(k)} | \mathbf{h}_i^{(k)})$, which randomly sets a fraction of the elements of $\mathbf{h}_i^{(k)}$ to 0. The encoder transforms the corrupted $\tilde{\mathbf{h}}_i^{(k)}$ into an abstract representation $\hat{\mathbf{h}}_i^{(k+1)}$, which is then used by the decoder to recover the uncorrupted $\mathbf{h}_i^{(k)}$. This process equips the DAE with the capability of extracting useful information for denoising, which is crucial for healthcare analytics, due to missing data and noise in real-world EMR data [26].

**Encoder of the $k$-th DAE.** The encoder of the $k$-th DAE transforms the corrupted representation using an affine transformation followed by a non-linear activation function:

$$\hat{\mathbf{h}}_i^{(k+1)} = f_\theta^{(k+1)}(\tilde{\mathbf{h}}_i^{(k)}) = \sigma(\mathbf{W}_\theta^{(k+1)} \tilde{\mathbf{h}}_i^{(k)} + \mathbf{b}_\theta^{(k+1)}) \tag{3}$$

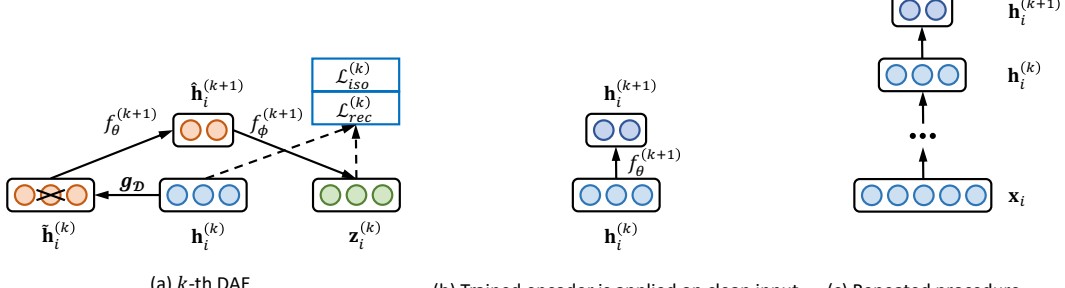

Figure 2: Model architecture of SDAE-based manifold learning.

where $f_\theta^{(k+1)}(\cdot)$ is the encoder with $\mathbf{W}_\theta^{(k+1)}$ and $\mathbf{b}_\theta^{(k+1)}$ as the weight matrix and bias vector, respectively. The rectified linear unit (ReLU) activation function $\sigma(\cdot)$ is used for non-linearity.

**Decoder of the $k$-th DAE.** The derived abstract representation $\hat{\mathbf{h}}_i^{(k+1)}$ is subsequently mapped back to the original space in the decoder, with the aim of recovering the uncorrupted representation:

$$\mathbf{z}_i^{(k)} = f_\phi^{(k+1)}(\hat{\mathbf{h}}_i^{(k+1)}) = \sigma(\mathbf{W}_\phi^{(k+1)}\hat{\mathbf{h}}_i^{(k+1)} + \mathbf{b}_\phi^{(k+1)}) \tag{4}$$

where $f_\phi^{(k+1)}(\cdot)$ is the decoder of the $k$-th DAE, with $\mathbf{W}_\phi^{(k+1)}$, $\mathbf{b}_\phi^{(k+1)}$ and the ReLU activation.

**Structure Preservation.** To attain a stable and robust abstract representation that is resilient to data corruption, it is crucial to recover the uncorrupted representation as accurately as possible. To achieve this, we adopt a reconstruction loss that preserves the data structure information. Given a batch of negative samples $\mathcal{B}$, the reconstruction loss for this batch is:

$$\mathcal{L}_{rec}^{(k)} = \sum_{i \in \mathcal{B}} \|\mathbf{h}_i^{(k)} - \mathbf{z}_i^{(k)}\|^2 \tag{5}$$

**Isotropy Constraint.** In addition to the reconstruction loss, it is essential to enforce an isotropy constraint to ensure that data Shapley value changes are uniform across orientations. To achieve this, we introduce a penalty that accounts for the change in data Shapley values relative to the Euclidean distance between two samples:

$$\mathcal{L}_{iso}^{(k)} = \sum_{i,j \in \mathcal{B}} \left(\frac{s_j - s_i}{\mu_{ij}}\right)^2 \tag{6}$$

where $i, j$ are two samples with $s_i, s_j$ as their data Shapley values, $\mu_{ij}$ as the distance between $\hat{\mathbf{h}}_i^{(k+1)}$ and $\hat{\mathbf{h}}_j^{(k+1)}$ derived from the encoder. The overall loss is then a weighted sum of the reconstruction loss and the isotropy penalty, jointly integrating the structural information and the isotropy information:

$$\mathcal{L}^{(k)} = -\frac{1}{|\mathcal{B}|}(\omega_{rec}\mathcal{L}_{rec}^{(k)} + \omega_{iso}\mathcal{L}_{iso}^{(k)}) \tag{7}$$

The weights $\omega_{rec}$ and $\omega_{iso}$ are introduced to address the issue of the two loss terms being on different scales. This ensures that both losses are decreased at similar rates, leading to a better balance between the optimization objectives [14, 29]. Specifically, the weights are set to the ratio between the respective loss in the current iteration $(t)$ and the loss in the previous iteration $(t-1)$:

$$\omega_{rec} = \mathcal{L}_{rec}^{(k)}(t)/\mathcal{L}_{rec}^{(k)}(t-1), \quad \omega_{iso} = \mathcal{L}_{iso}^{(k)}(t)/\mathcal{L}_{iso}^{(k)}(t-1) \tag{8}$$

We have introduced how to learn the $k$-th DAE using the loss function in Equation 7, as shown in Figure 2(a). The corrupted input is only used during the initial training to learn robust feature extractors. After the encoder $f_\theta^{(k+1)}(\cdot)$ is trained, it will be applied to the clean input as in Figure 2(b):

$$\mathbf{h}_i^{(k+1)} = f_\theta^{(k+1)}(\mathbf{h}_i^{(k)}) = \sigma(\mathbf{W}_\theta^{(k+1)}\mathbf{h}_i^{(k)} + \mathbf{b}_\theta^{(k+1)}) \tag{9}$$

$\mathbf{h}_i^{(k+1)}$ will be used as input for the $(k+1)$-th DAE, as in Figure 2(c), to continue the repeated training process. When the last DAE, i.e., $(K-1)$-th DAE, is trained, we obtain the encoded representation $\mathbf{h}_i^{(K)}$ in the manifold space $\mathcal{S}'$, which preserves the data structure information in $\mathbf{x}_i$ and integrates the desired isotropy constraint. $\mathbf{h}_i^{(K)}$ will serve as input for subsequent medical cohort discovery.

### 3.3 Cohort Discovery Among High Data Shapley Value Negative Samples

We proceed to perform cohort discovery in the encoded manifold space $\mathcal{S}'$, where each negative sample's input $\mathbf{x}_i$ is transformed into $\mathbf{h}_i^{(K)}$. We begin by setting a threshold value $\tau$ to filter out negative samples with data Shapley values below $\tau$, which focuses our analysis on negative samples with high data Shapley values, i.e., high contributions to the prediction task. Among the remaining negative samples with high data Shapley values, we target to detect the hot zones in $\mathcal{S}'$, which may represent medically meaningful cohorts of arbitrary shape.

To achieve this, we employ DBSCAN, short for density-based spatial clustering of applications with noise [9, 10, 39] on such samples. The core idea of DBSCAN is to group samples that are close to each other in the manifold space $\mathcal{S}'$ into clusters, which could locate potential cohorts, whereas treating the remaining samples as noise or outliers. DBSCAN has three main steps: (i) identify the points within each point's $\varepsilon$- neighborhood and determine the "core points" with over $P_{min}$ neighbors; (ii) detect the connected components of the core points in the neighbor graph, disregarding any non-core points; (iii) assign each non-core point to the clusters which are the $\varepsilon$-neighborhood of the point; otherwise, label the point as noise. This process results in a set of clusters $\{C_1, C_2, \ldots, C_R\}$ and a set of noisy samples $\Psi$. Given the clusters, we define cohorts as follows.

**Definition 2** *(Cohorts) For a dense cluster $C_r$ identified by the DBSCAN algorithm, we consider each of its core points and define a spherical space with the core point as its center and $\varepsilon$ as its radius. The joint space of all such spherical spaces is the cohort we aim to discover from this cluster.*

These discovered cohorts provide a promising avenue for further exploration of medically meaningful patterns in EMR data analytics, potentially revealing important insights.

## 4   Experimental Evaluation

We evaluate our proposal using our hospital's EMR data, on which we utilize 709 lab tests to predict whether a patient will develop AKI in each admission in two days (as defined in Section 2). In total, we receive 20,732 admissions, of which 911 develop AKI. We partition the dataset into training data (90%) and testing data (10%). We employ the logistic regression model to compute the data Shapley value for each negative sample as detailed in Section 3.1, using the area under the ROC curve (AUC) as the evaluation metric, and perform Monte Carlo permutation sampling 100,000 times with early stopping. For the manifold learning step, we utilize an SDAE comprising 3 DAEs. The 709-dimension inputs are transformed using encoders with dimensions 256, 128, and 64, respectively.

### 4.1   Cohort Discovery in Clinical Validation

We present the cohort discovery results on our dataset in Figure 3, where we first display the data Shapley value histogram among all the negative samples in Figure 3(a). It is noteworthy that this histogram can be well fitted by a Gaussian mixture model, consisting of three distinct and interesting components. We next examine each component in detail. The first component on the left represents the negative samples with negative data Shapley values. These samples have a negative impact on the prediction task, meaning that they are detrimental to predicting the AKI occurrence. In prior studies, one generally plausible explanation for the presence of such samples is the existence of mislabeled data [12]. However, for a representative acute disease like AKI, these negative samples are highly likely to be positive samples in the future but have not yet exhibited symptoms of AKI within the monitored time duration. Moving on to the second component in the middle, we observe that its data Shapley values are centered around a mean value close to zero. This implies that these negative samples are generally healthy without any apparent AKI-related symptoms. Notably, these healthy samples constitute a relatively significant portion of the data, which is commonly observed in clinical practice and aligns with our initial expectations. The third component on the right represents negative samples that are particularly valuable for the prediction task and merit special attention in our study. To further investigate these samples, we introduce a separation line between the second and third components, i.e., a threshold 60% to exclude the lower 60% negative samples based on their data Shapley values while retaining the remaining 40% for further analysis. Our focus is on these remaining 40% samples for identifying the hot zones, as illustrated in Figure 1.

The distribution of all negative samples, in terms of their data Shapley values in the manifold space, is presented in Figure 3(b). Upon performing DBSCAN on the extracted 40% samples with high data Shapley values (points brighter than dark blue), we identify seven distinct cohorts of interest,

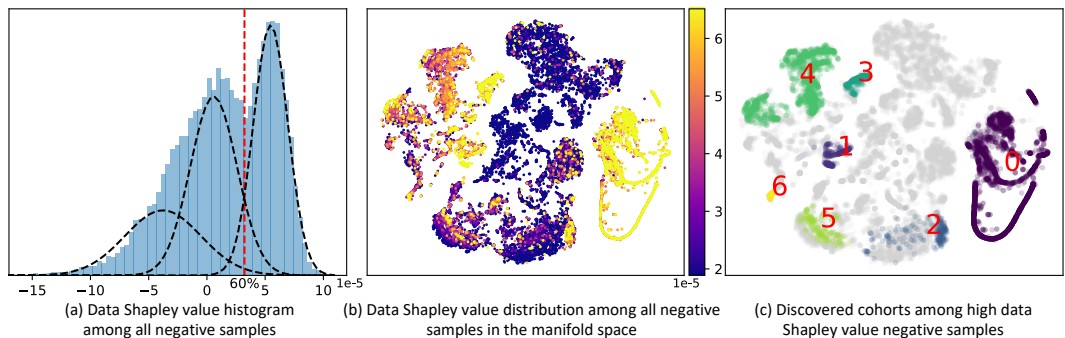

(a) Data Shapley value histogram among all negative samples

(b) Data Shapley value distribution among all negative samples in the manifold space

(c) Discovered cohorts among high data Shapley value negative samples

Figure 3: Cohort discovery on our dataset.

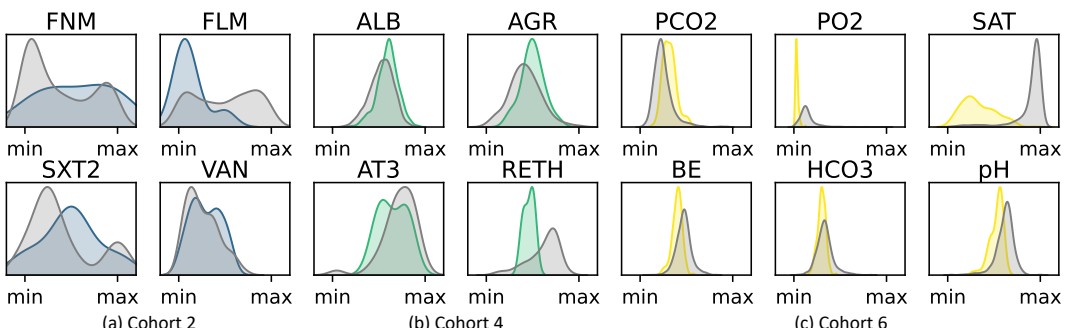

(a) Cohort 2

(b) Cohort 4

(c) Cohort 6

Figure 4: Lab test patterns of discovered Cohorts 2, 4, and 6. In each cohort, the colored region (blue, green, and yellow) represents the lab test value probability density of the samples in the cohort, while the grey region denotes that of all the other samples outside the cohort.

which are visually displayed using t-SNE plots in Figure 3(c), in which grey points are either with low data Shapley values or labeled as noise by DBSCAN. We observe that these discovered cohorts are distinguishable from one another, potentially corresponding to medically meaningful patterns.

## 4.2 In-depth Analysis of Discovered Cohorts

**Cohort 2: inflammatory cohort.** Figure 4(a) indicates a pronounced neutrophil-to-lymphocyte ratio (NLR) [48] in this patient group, marked by an increase in neutrophils (FNM) and a decrease in lymphocytes (FLM). This pattern, often tied to infectious, inflammatory, and stress conditions, suggests an overactive immune response leading to reduced lymphocyte counts [36, 8]. An elevated NLR, a reliable inflammatory marker, indicates a propensity for invasive infections [16]. Meanwhile, the levels of Cotrimoxazole (SXT2) and Vancomycin (VAN), both administered to treat infections including those associated with methicillin-resistant staphylococcus [15], are found to be elevated in the bodies of these patients. The findings suggest that this patient cohort comprises individuals experiencing infections and acute inflammation, and receiving antibiotic treatment. Severe infections can cause systemic inflammatory response syndrome and kidney injury. Antibiotics like vancomycin can worsen kidney stress and have nephrotoxic properties [47], potentially leading to kidney dysfunction during treatment. However, modern medical practice can effectively manage these cases. Infections are promptly treated with broad-spectrum antibiotics and at appropriate doses within safety limits; hence, the patients do not develop significant AKI [13].

**Cohort 4: hepatic and hematological disorders cohort.** As delineated in Figure 3(c), Cohort 4 exhibits an augmented region and an increased quantity of sampling points, indicative of a more expansive patient population. A comprehensive analysis of the lab test indicator distribution for this cohort, portrayed in Figure 4(b), reveals differences in levels of serum proteins. Specifically, derangements in levels of albumin (ALB) and the albumin-globulin ratio (AGR) signify aberrant protein synthesis in patients. These may be associated with hepatic dysfunction or hematological diseases such as myeloma [41, 27]. Hepatic diseases can lead to impaired production of other proteins such as antithrombin III (AT3) [21]; AT3 may also be lost excessively in nephrotic syndrome

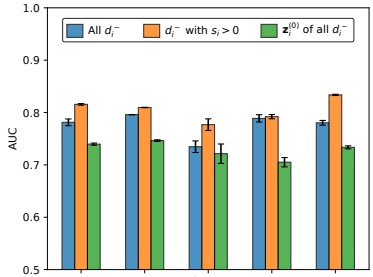
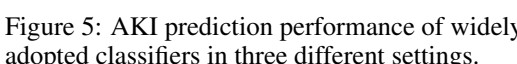
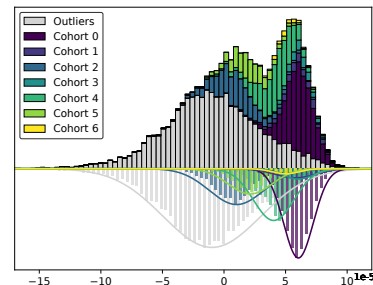

Figure 5: AKI prediction performance of widely adopted classifiers in three different settings.

Figure 6: Data Shapley value histogram of the samples within our discovered cohorts.

which is a kidney disorder [18], or undergo accelerated consumption in disseminated intravascular coagulation [34]. Diminished reticulocyte hemoglobin (RETH) is associated with iron deficiency anemia [2], and could either be linked to hematological disorders or nutritional deficiency. In addition, imbalances in albumin and globulin may also be associated with dehydration. Therefore, our observation derived from Cohort 4 may support the pathophysiological relationship that exists between disorders of the hematological and hepatic systems, which increases the propensity for kidney disease. Clinicians should exercise vigilance in care when managing these cases.

**Cohort 6: respiratory failure and metabolic acidosis cohort.** Figure 4(c) reveals significant metabolic imbalances in patients, leading to an acid-base imbalance. Specifically, increased carbon dioxide pressure (PCO2), reduced oxygen pressure (PO2), and insufficient blood oxygen saturation (SAT) suggest respiratory failure [5]. Concurrently, reduced base excess (BE), bicarbonate ion (HCO3) levels, and blood pH values hint at metabolic acidosis, indicating possible acute illnesses causing lactic or ketoacidosis [24]. These results suggest potential severe respiratory complications, such as advanced pneumonia, heart failure-induced pulmonary edema, or chronic obstructive pulmonary disease (COPD)[20]. Alternatively, acute conditions like hypoxia, shock, or severe infection could disrupt aerobic metabolism, leading to anaerobic glucose conversion to lactate, which accumulates in the bloodstream and causes acidosis. This puts significant strain on the kidneys, potentially resulting in renal disease symptoms[25]. This cohort of patients under examination does not advance to AKI, leading to the inference that renal dysfunction may not constitute an end-organ complication Rather, this patient cohort appears to exhibit a heightened disposition to respiratory failure.

### 4.3 Validation of Effectiveness for Each Component

We validate the effectiveness of each component in our approach for AKI prediction. Specifically, we evaluate three settings of the negative sample usage in the training data (with positive samples the same): (i) all $d_i^-$: use all negative samples; (ii) $d_i^-$ with $s_i > 0$: only use the negative samples with positive data Shapley values; (iii) $\mathbf{z}_i^{(0)}$ of all $d_i^-$: use the decoded representations from the SDAE-based manifold learning. $\mathbf{z}_i^{(0)}$ is in the same dimension as the raw input but is in the decoding space after transformation by SDAE. To ensure the credibility of our conclusions across different settings, we evaluate several widely adopted classifiers: logistic regression (LR), gradient-boosting decision tree (GBDT), adaptive boosting (AdaBoost), random forest (RF), and multilayer perceptron (MLP). The experimental results in AUC (mean $\pm$ std) from five repeats are illustrated in Figure 5.

**Effectiveness of the Negative Sample Shapley Field.** By comparing settings (i) and (ii), we explore the effectiveness of our constructed Negative Sample Shapley Field. The results clearly demonstrate that by removing negative samples with data Shapley values smaller than $0$, all the classifiers exhibit an improvement in AUC. This finding supports the rationale behind our approach of linking samples of great medical concern with their data Shapley values. Additionally, the effectiveness of approximating data Shapley values through Monte Carlo permutation sampling is further validated. Thus, this confirms the efficacy of our constructed Negative Sample Shapley Field.

**Effectiveness of Manifold Learning.** By changing the input data from the raw space to the decoder's output space after our proposed SDAE-based manifold learning (settings (i) vs. (iii)), we observe a moderate decrease in AUC, approximately $5\%$ in most classifiers. This decrease aligns with our expectations, as the transformation in SDAE introduces a certain level of information loss. However,

the performance degradation remains within an acceptable range. These findings demonstrate that our proposed manifold learning manages to preserve the original data structure information and effectively model the original raw data space, despite a significant reduction in data dimension from 709 to 64. Thus, this corroborates our design rationale of employing SDAE for manifold learning with structure preservation and isotropy constraint.

**Effectiveness of Cohort Discovery.** We further validate our method's ability to decompose high data Shapley value samples into distinct, medically relevant cohorts. Figure 6 presents the data Shapley value histogram of our identified cohorts, with the upper part aligned with Figure 3(a) but color-coded by cohort proportion. The lower part shows each cohort's data Shapley value distribution. We note seven cohorts effectively partition Figure 3(a)'s third component into Gaussian distributions, implying consistent data Shapley values within each cohort. Cohort 2, identified as the inflammatory group, exhibits relatively lower data Shapley values, as immune abnormalities cannot serve as specific features for kidney injury. Conversely, Cohorts 4 and 6, involving critical metabolic systems, display higher data Shapley values, which indicates their significant medical relevance to AKI prediction. These observations confirm the homogeneity within each cohort due to DBSCAN's detection capability, and similarity in data Shapley values, further substantiating our proposed isotropy constraint in manifold learning. In essence, our approach effectively identifies proximate cohorts with similar data Shapley values, providing valuable medical insights for the prediction task.

## 5  Related Work

Shapley value, originally introduced in cooperative game theory [40], offers a solution for the equitable distribution of a team's collective value among its individual members [7]. Notable applications of the Shapley value in machine learning encompass data valuation, feature selection, explainable machine learning, etc [38, 12, 46, 31, 31, 30]. Among these applications, data valuation holds particular significance in quantifying the contributions of individual data samples toward predictive models. In this research line, the data Shapley value [12] presents an equitable valuation framework for data value quantification with subsequent research focusing on enhancing computational efficiency [17, 11].

Representation learning is a crucial research area contributing to the success of many machine learning algorithms [3]. Among the representation learning methods, manifold learning stands out due to its capability of reducing the dimensionality and visualizing the underlying structure of the data. Traditional manifold learning methods include Isomap [42], locally linear embedding [37], and multi-dimensional scaling [4]. In recent years, AEs have gained significant attention in representation learning, offering efficient and effective representations of unlabeled data. Researchers develop various AE variants for specific application scenarios, e.g., regularized AEs [1], sparse AEs [33], DAEs (denoising AEs) [43]. Specifically, DAEs and their advanced stacked variant SDAEs [44] are highly suitable to tackle EMR data, in which missing and noisy data remains a notorious issue [26].

DBSCAN, short for density-based spatial clustering of applications with noise, is introduced to alleviate the burden of parameter selection for users, facilitate the discovery of arbitrarily-shaped clusters, and demonstrate satisfactory efficiency when dealing with large datasets [9, 10, 39].

## 6  Conclusion

This paper proposes to examine negative samples for cohort discovery in healthcare analytics, which has not been explored in prior research. In particular, we propose to measure each negative sample's contribution to the prediction task via its data Shapley value and construct the Negative Sample Shapley Field to model the distribution of all negative samples. To enhance the cohort discovery quality, we transform this original field into an embedded space using manifold learning, incorporating the original data structure information and isotropy constraint. In the transformed space, we manage to identify medically meaningful cohorts within negative samples by DBSCAN. The experiments on our hospital's EMR data empirically demonstrate the effectiveness of our proposal, and the medical insights derived from our discovered cohorts are validated by clinicians, highlighting the medical value of our approach. Future work includes conducting a long-term validation to further verify the conclusions drawn from cohort discovery. Additionally, more detailed analyses and fine-grained clinical validation are required to explore the detected cohorts that exhibit a hierarchical structure.

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
