# OpenReview forum: "Exploiting Negative Samples: A Catalyst for Cohort Discovery in Healthcare Analytics"
_NeurIPS.cc/2023/Conference — Submitted to NeurIPS 2023_

### Official Review · Reviewer_Sxir · 2023-07-07

**Soundness:** 2 fair
**Presentation:** 3 good
**Contribution:** 2 fair
**Rating:** 3
**Confidence:** 5

**Summary:**

This paper presents a Shapley value based cohort discovery, by constructing "Negative Sample Shapley Field" that possesses isotropy property. By doing so, negative samples can be effectively clustered and separated with respect to the Shapley values.

**Strengths:**

I think this paper points out many important problems in healthcare research. Specifically,
1: how to deal with pos/neg imbalance and how to make better use of vast negative samples?
2: how to identify negative samples that are more useful for the current research problem?

The use of latent variable models to deal with misssingness in EHR dataset is a promising approach too.


**Weaknesses:**

1: Correct me if I'm wrong, but I think Eq.2 is ill-defined. For any metric M, let's say accuracy, then $s_i = s_j$ as long as both negative samples have the same predicted labels by a predictor F, right?
2: Many parts need justifications. For example, a) why is the defined metric in Eq.2 means high contribution to prediction task? b) why does Eq.6 help with isotropy? c) How is k-th DAE different from a normal encoder with k layers?
3: I don't see why DBSCAN + AE is proposed as a contribution when you can simply use VAE.
4: I doublt the logic between line 248 and line 249. The defined Shapley value in Eq.2 is 0 does not mean these patients are healthy. Note that you are defining M=AUROC, therefore M has a very stable value when you have sufficient samples to draw a smooth ROC curve when calculating $M(D^+ \cup A)$. As a result, $s_i = E[ M(D^+ \cup A \cup d_i) - M(D^+ \cup A)]$ is usually zero.


**Questions:**

1: Line 100: Why is Shapley field continuously differentiable?
2: Why Fig 1.b misdicovered hot zones? The zone bounded by these 4 points is the "hot zone". What is the definition of misdiscovery?
3: Line 178-181: How is the k-th DAE different from a vanilla encoder with k layers?
4: Line 183: I think you are referring to the previous latent space rather than "original space".
5: How is the $L_{iso}$ contributing to isotropy? It is simply maximizing the average Euclidean distances of two samples between two latent spaces.
6: Line 211: why is high value in Eq.2 leads to high contribution to the prediction task?

---

> ### Author Rebuttal · Authors · 2023-08-09
>
> We would like to express our gratitude for your detailed comments regarding our paper and respond to them point by point in detail below.
>
> **W1.**
> In Eq.1, given a metric $M$ such as accuracy, two negative samples are assigned the same data Shapley value ($s_i = s_j$) if and only if these two samples bring the same marginal contributions to the model performance **when added to all subsets of negative samples**. Following this, we introduce Eq.2 to approximate the computation of data Shapley values in Eq.1 through Monte Carlo permutation sampling. In Eq.2, $s_i = s_j$ is achieved if and only if these two negative samples exhibit the same expectation of marginal contributions to the model performance **for all the Monte Carlo permutations**.
>
> **W2a).**
> Shapley values are widely adopted in cooperative game theory for transferable utility cooperative games. In such games, there are a player set and a characteristic function that assigns values to subsets of players, and Shapley values are widely acknowledged to provide a rigorous and intuitive way to distribute the collective value of the team fairly across individual players [38]. Grounded on Shapley values, data Shapley values tackle the cooperative game of data valuation, where the player set represents data samples, and the characteristic function is defined as the model's payoff, typically measured by goodness-of-fit metrics such as accuracy or AUC [12].
>
> In this work, we define data Shapley values for negative samples in Eq.1 (Proposition 1) and propose an approximation method using Monte Carlo permutation sampling in Eq.2. After demonstrating the validity of Proposition 1, we note that our defined Eq.2 is able to quantify the contribution of each negative sample to the model's performance in a prediction task.
>
> **W2b).**
> In Eq.6, $s_j-s_i$ denotes the difference in data Shapley values between two negative samples, and $\mu_{ij}$ represents the Euclidean distance between these two samples. Therefore, Eq.6 introduces a penalty term that considers the change in data Shapley values relative to the Euclidean distance for all pairs of samples. By doing so, this equation promotes uniform changes in data Shapley values across all orientations, consequently contributing to isotropy.
>
> **W2c).**
> We highlight that denoising autoencoders (DAE) distinguish themselves from standard autoencoders by their unique capability of **cleaning partially corrupted input, or in short denoising** enabled by their reconstruction criterion. The proposal of DAE, and further stacked denoising autoencoders (SDAE), is based on the notion that "a good representation is one that can be obtained robustly from a corrupted input and that will be useful for recovering the corresponding clean input" [43, 44]. The denoising capability of DAE holds crucial significance for healthcare analytics on account of missing data and noise in real-world EMR data [26].
>
> **W3.**
> We employ SDAE to preserve structure in data and enforce an isotropy constraint during manifold learning, and then conduct spatial clustering via DBSCAN in the transformed space for cohort discovery. However, variational autoencoders (VAE) are probabilistic generative models which are generally adopted in applications such as image generation, image style transfer, and anomaly detection rather than spatial clustering. Hence, we are not clear how VAE could be a promising alternative for achieving our goal of cohort discovery in this work. We would appreciate if more details regarding this comment could be provided.
>
> **W4.**
> We note that this doubt originates from the interpretation of our defined data Shapley values and the assumption that "M has a very stable value **when you have sufficient samples** to draw a smooth ROC curve when calculating $M(\mathcal{D}^+ \cup A)$" does not hold.
>
> We define the data Shapley values for negative samples in Eq.1 and further propose an approximation by Monte Carlo permutation sampling in Eq.2. A negative sample is assigned a data Shapley value of 0 in Eq.1 if and only if it consistently brings no marginal contribution to the model performance **when added to all subsets of negative samples**. Similarly, a negative sample exhibits a 0 data Shapley value in Eq.2 if and only if its expectation of marginal contributions to the model performance is 0 **across all the Monte Carlo permutations**. Thus, negative samples with the defined Shapley values of 0 in Eq.2 provide no contribution to the prediction task, i.e., hospital-acquired AKI prediction, indicating that these samples are "generally healthy without any apparent AKI-related symptoms" and lack informative insights for the task.
>
> **Q1.**
> Our proposed Negative Sample Shapley Field is a scalar field depicting the relationship between negative samples and their respective data Shapley values. With the data Shapley values for negative samples defined in Eq.1, as a performance metric function of a continuously differentiable prediction model (in our case, logistic regression), $M$ is continuously differentiable (in our case, AUC). Therefore, the data Shapley values in Eq.1, in the form of a linear combination of $M$ functions, are continuously differentiable as well.
>
> **Q2.**
> In Figure 1(b), density-based spatial clustering assigns the 4 points into two different hot zones, which renders the actual hot zone missed.
> To address such mis-discovery leading to ineffective cohort discovery, we propose an isotropy constraint, the impact of which is investigated in our response to Weakness 2 of Reviewer g4r8's reviews.
>
> **Q3/5/6.**
> Our detailed explanations for answering these three questions can be found in our responses to Weaknesses 2c), 2b), and 2a) of your reviews.
>
> **Q4.**
> Yes, we are referring to the previous latent space to which the decoder of the $k$-th DAE transforms the abstract representation. Thanks for pointing out this issue, and we will revise this statement for conciseness.

---

> > ### Comment · Reviewer_Sxir · 2023-08-16
> > **Thanks for the response**
> >
> > Thanks for the response. I would like to further discuss my question about $W1$: when calculating the "marginal contributions of the model performance", do you retrain a huge set of models {F1, F2, ...} during MC sampling (one model for each subset of negative samples)?
> > If you don't, you will see that they are equal as long as both negative samples have the same predicted labels by the same predictor F.
> > if you do: first of all, I would suggest that you use a different notation (perhaps bold or smaller case) because F is changing as a functioal variable. Now, let's take the simplest form of F, a linear SVM. Note that F will only change if the new sample s is a support vector, otherwise F stays the same. In that sense, s_j and s_i are equal as long as both negative samples have the same predicted labels and they are not the support vectors. Note that in machine learning scenario, most of the samples are not support vectors, so they will have same Shapley value. Is that true?

---

> > > ### Author Response · Authors · 2023-08-18
> > > **Thanks for your reply**
> > >
> > > Thanks for your reply and further discussions. To clarify our approach, we start by presenting a concrete example elucidating how Shapley values are computed in a cooperative game theory.
> > >
> > > Imagine three friends, A, B, and C, pondering over how to split their dining bill based on their individual and combined appetites. They realize that the total cost doesn't simply equate to the sum of individual costs. Here's their observation:
> > >
> > > ```
> > > Alone: A = $7; B = $4; C = $6
> > > In pairs: A & B = $7; A & C = $15; B & C = $9
> > > All together: A, B, & C = $19
> > > ```
> > >
> > > To equitably distribute the bill, they look at every possible sequence (i.e., permutation) in which they could have entered the restaurant and ordered (i.e., adding the samples one at a time into the training dataset), and they can get their "bill increment" of each permutation (i.e., marginal contribution):
> > >
> > > ```
> > > A($7) C($8) B($4)
> > > A($7) B($0) C($12)
> > > B($4) C($5) A($10)
> > > B($4) A($3) C($12)
> > > C($6) A($9) B($4)
> > > C($6) B($3) A($10)
> > > ```
> > >
> > > From these sequences and the bill amounts, the Shapley value for each friend is:
> > >
> > > - A = (7 + 7 + 10 + 3 + 9 + 10)/6 = 7.67
> > > - B = (4 + 0 + 4 + 4 + 4 + 3)/6 = 3.17
> > > - C = (8 + 12 + 5 + 12 + 6 + 6)/6 = 8.17
> > >
> > > Thus, based on the Shapley values, A should contribute about 7.67, B should contribute 3.17, and C should contribute 8.17 to the bill, if they have a meal together.
> > >
> > > In the context of data Shapley values, each individual data sample assumes the role of a "friend," and the model's performance metric (such as accuracy or AUC) when utilizing a specific subset of data samples as the training set is equated to the "bill amount." Due to the inherent exponential complexity involved in calculating data Shapley values, the Monte Carlo permutation sampling method has gained widespread acceptance as an approximation technique [12, 38]. That is to say, it is not feasible to evaluate "every possible sequence" at factorial-level quantity, but instead, uniformly sampling a certain number of sequences is deemed reasonable. We hope this concrete example could help elucidate the process of computing data Shapley values for negative samples.
> > >
> > > To address your specific queries:
> > >
> > > **1. Retraining during Monte Carlo permutation sampling:**
> > > Yes, we need to retrain the model $F$ during Monte Carlo permutation sampling. Specifically, in each permutation, we add the negative samples one by one and for each added negative sample, we need to retrain $F$ to calculate its contribution to the current permutation. The data Shapley value of a negative sample is the aggregation of its marginal contribution across permutations. We have included the detailed descriptions of this process in Section B.2 of the Appendices, with the pseudo-code in Algorithm 1. Thanks for your suggestion on notations and we will update the notation of $F$ as advised.
> > >
> > > **2. When utilizing linear SVM as $F$:**
> > > While the assertion "$F$ will only change if the new sample $s$ is a support vector, otherwise $F$ stays the same" is correct, it is pivotal to understand that the calculation of the data Shapley value for the sample $s$ is based on all permutations (as shown in the example above). Within each permutation, **whether $s$ is a support vector is not its inherent property**, but is dependent on whether $s$ is a sample for determining the max-margin hyperplane **when $s$ is added to the training set**. In other words, whether $s$ is a support vector is related to its preceding samples in the permutation. For the current permutation, if the added sample $s$ becomes a support vector, then $F$ will be changed; otherwise, $F$ will stay the same. Further, for each sample $s_i$, it will always become a support vector in certain permutations, e.g., when it is added to the permutation as the first sample, resulting in influences/changes on $F$. Consequently, given two different samples $s_i$ and $s_j$, their influences/changes on $F$ will be different.
> > >
> > > Finally, we would like to highlight that linear SVM is a special case of prediction models in that $F$ is solely determined by support vectors. In most prediction models, we generally aim to minimize a loss function in which all the samples in the training set are involved, so the introduction of a new sample modifies the loss function, and consequently, $F$.
> > >
> > > We hope this elaboration addresses your queries. If any aspect remains ambiguous or warrants further discussion, please do not hesitate to reach out.

---

> > > > ### Comment · Reviewer_Sxir · 2023-08-18
> > > >
> > > > Thank you for clarifying that F has to be retrained for each permutation and for each calculation of a negative data. In that case, I am very concerned about the feasibility of this approach. As you have mentioned, a linear SVM is a special case and in most cases you need to re-train with the current permutation. Not to consider the LLMs that takes weeks for training, even a small neural network will be impossible considering the number of permutations you want to take and the number of samples you want to calculate, not to say the storage and loading of so many trained models.

---

> > > > > ### Author Response · Authors · 2023-08-18
> > > > > **Thanks for your reply**
> > > > >
> > > > > Thanks for your reply.
> > > > >
> > > > > We note that you gave Reject for our paper. However, as of now, all the points listed in Weaknesses and Questions of your reviews are solely questions on techniques of our proposal, rather than identifying weaknesses or suggesting improvements for the manuscript itself.
> > > > >
> > > > > We would like to verify if all your inquiries have been adequately addressed in our responses. If that is the case, we can shift our focus toward discussing the concern you newly raised regarding feasibility.
> > > > >
> > > > > Respectfully, we disagree with the assertion that our proposal is constrained in terms of feasibility.
> > > > >
> > > > > Firstly, Shapley values are widely recognized techniques for quantifying the contributions of participants across various domains, including machine learning and deep learning [38]. Numerous successful applications exist that exemplify the practicality of Shapley values.
> > > > >
> > > > > Secondly, the substantial computational complexity associated with calculating Shapley values has spurred the development of different optimization techniques to enhance efficiency, such as Monte Carlo permutation sampling and truncation adopted in our proposal (details in Section B.2 of the Appendices). It's important to note that our focus is orthogonal to these optimizations, as we are dedicated to constructing the Negative Sample Shapley Field based on calculated Shapley values for cohort discovery. Therefore, if novel optimization techniques are introduced in the future, they would directly benefit our proposal.
> > > > >
> > > > > Thirdly, we feel compelled to highlight that your ongoing discussion considerably deviates from the core focus of our work - LLMs are too far away from our research scope. Our research is centered around Electronic Medical Records (EMR) collected from patients' hospital visits. The dataset we have employed is notably extensive within the realm of EMR analytics. Our evaluation has substantiated the effectiveness of our proposal, yielding insightful medically validated results. Consequently, the feasibility of our proposal has been effectively confirmed within the scope of its intended task.
> > > > >
> > > > > In conclusion, we earnestly request that you reconsider our work in light of the clarifications provided in response to your reviews. Should any further inquiries for clarification persist, please don't hesitate to inform us. We genuinely appreciate your time and consideration.
> > > > >
> > > > > [38] Benedek Rozemberczki, Lauren Watson, Péter Bayer, Hao-Tsung Yang, Oliver Kiss, Sebastian Nilsson, and Rik Sarkar. The shapley value in machine learning. In IJCAI, pages 5572–5579. ijcai.org, 2022.

---

> > > > > > ### Comment · Reviewer_Sxir · 2023-08-18
> > > > > >
> > > > > > I would like to thank the authors for the reply. As the authors feel that the discussions are off topic, I would like to have my last comment here.
> > > > > > 1: I don't see why the concerns about the techniques in the proposed work are not the weakness of this paper.
> > > > > > 2: Whether the cited paper or the actual Shapley-based ML paper (SHAP: https://arxiv.org/abs/1705.07874) are talking about perturbing the datasets with a fixed model so that Shapley estimation is feasible. As far as I know, none of the recent Shapley based methods require retraining of models with respect to the permutation of data.
> > > > > > 3: I doublt that large machine learning models are far away from EMR dataset, considering many clinical BERT models (and diffusion models, which are pretty slow in training too) are recently proposed using MIMIC and Physionet dataset. Even so, simply considering the used dataset in this paper and the appendix B.2: in the extreme scenario that maximally simplify everything with a LR and a huge truncation_tolerance such that the loop stops immediately, then there will still be at least 2 well-trained LR models for each negative sample. Considering the dataset has 20K negative patients, there will be at least 40K well-trained models. I don't think this is feasible in practice.

---

> > > > > > > ### Author Response · Authors · 2023-08-18
> > > > > > > **Thanks for your reply**
> > > > > > >
> > > > > > > Thanks for your reply.
> > > > > > >
> > > > > > > **Response to 1.** We aim to provide comprehensive responses to your reviews and address your concerns during the response phase. Since the points raised in Weaknesses and Questions of your reviews are all in the form of questions, our responses focus on addressing these questions to the best of our ability. Therefore, we would like to confirm whether all of your inquiries have been adequately addressed in our responses. Based on your latest reply, we believe that all of your inquiries have indeed been addressed.
> > > > > > >
> > > > > > > **Response to 2.** We would like to point out that your statement, "none of the recent Shapley based methods require retraining of models with respect to the permutation of data," is **wrong**.
> > > > > > >
> > > > > > > In the survey paper we cited [38], particularly in **Section 4.2 Data Valuation**, a review of representative studies utilizing Shapley values to quantify the contributions of data samples to prediction tasks reveals that their computation necessitates the retraining of models concerning data permutations. Examples of such studies include [12], [17], and [11]. Therefore, this approach is widely recognized within the community.
> > > > > > >
> > > > > > > Regarding the SHAP paper you mentioned, it falls into a distinct category of utilizing Shapley values for explainability, as outlined in **Section 4.4 Explainable Machine Learning** of the paper we cited [38]. This is less relevant to our study.
> > > > > > >
> > > > > > > **Response to 3.** Fundamentally, the computation of data Shapley values is merely one component of our study. Our contributions do not revolve around the methodology for obtaining data Shapley values.
> > > > > > >
> > > > > > > We understand that your rejection is solely based on your criticism about the feasibility of computing data Shapley values, which suggests that such a technique is not practically viable, rendering all downstream studies dependent on data Shapley values not sound. We respect your personal opinion, despite the fact that this technique has received widespread recognition within the community.
> > > > > > >
> > > > > > > Our evaluation conducted for this paper involved a total of 100K Monte Carlo permutation sampling, with an average truncation length of 376. These computations were carried out on a dual-CPU server in under a week, which is a reasonable timeframe and cost for examining patient cohort characteristics, as addressed in our study.
> > > > > > >
> > > > > > > [38] Benedek Rozemberczki, Lauren Watson, Péter Bayer, Hao-Tsung Yang, Oliver Kiss, Sebastian Nilsson, and Rik Sarkar. The shapley value in machine learning. In IJCAI, pages 5572–5579. ijcai.org, 2022.
> > > > > > >
> > > > > > > [12] Amirata Ghorbani and James Y. Zou. Data shapley: Equitable valuation of data for machine learning. In ICML, volume 97 of Proceedings of Machine Learning Research, pages 2242–2251. PMLR, 2019.
> > > > > > >
> > > > > > > [17] Ruoxi Jia, David Dao, Boxin Wang, Frances Ann Hubis, Nick Hynes, Nezihe Merve Gürel, Bo Li, Ce Zhang, Dawn Song, and Costas J. Spanos. Towards efficient data valuation based on the shapley value. In AISTATS, volume 89 of Proceedings of Machine Learning Research, pages 1167–1176. PMLR, 2019.
> > > > > > >
> > > > > > > [11] Amirata Ghorbani, Michael P. Kim, and James Zou. A distributional framework for data valuation. In ICML, volume 119 of Proceedings of Machine Learning Research, pages 35353544. PMLR, 2020.

---

### Official Review · Reviewer_vvb4 · 2023-07-08

**Soundness:** 2 fair
**Presentation:** 2 fair
**Contribution:** 2 fair
**Rating:** 3
**Confidence:** 3

**Summary:**

The paper addresses the cohort discovery problem for supervised learning in the machine learning for healthcare domain. Positive examples of the cohort are easy to identify while it is not as straightforward to determine which negative examples should be admitted into a cohort.  To deal with this problem, the paper calculates the data Shapley value of the negative samples in the dataset. Then, the paper carried out representation learning using a stacked denoising autoencoder to mitigate the nonuniform changes of Shapley value in the original feature space. Finally, the paper carried out clustering in the learned representation space to identify important negative examples to create the cohort. The paper evaluated the proposed method on a clinical dataset to demonstrate the utility of the proposed method.

**Strengths:**

* The paper deals with the cohort discovery problem, which is an important problem in the machine learning for healthcare domain.

* The proposed method is straightforward and makes sense to me for the most part.

* In terms of empirical evaluation, the paper provides a detailed explanation of the cohorts discovered from a clinical perspective, although I won't be able to judge whether such findings make precise, clinical sense given that I do not have a medical background. I also appreciate the authors carried out additional experiments to study the effectiveness of each component of the proposed method.

**Weaknesses:**


* Because the proposed method is straightforward and directly takes advantage of existing methods, I am not quite sure whether the paper has enough technical novelty from a machine-learning perspective.

* Regarding experiments, while I think the authors dive deep into providing an analysis of the outcome of the proposed method from a clinical perspective, there are no alternative methods compared to the proposed method to understand the performance of the proposed method. It would also be interesting to see the proposed method applied to more than just one dataset as discussed in the paper. Finally, it should also be noticed that identifying relevant negative examples is not a problem that is exclusive to the healthcare domain. Many application domains will be interested in the proposed method to identify relevant negative examples for binary classification problems. As such, the authors may also consider applying their methods beyond the medical domain down the road.

* Clarity of the paper can be improved. Some key concepts are not well explained. For example, what is the role of data Shapley value? It appears to be the contribution of a data point to the learned classifier. The authors do not seem to elaborate on this concept enough in the paper. What's the intuition behind it? Why it makes sense to use Shapley value to measure contribution?  I also don't think the authors explain well the phenomenon of "the non-uniform distribution of negative samples with similar data Shapley values". Further intuition on this point will help to better motivate the need for representation learning.



**Questions:**

The author can better explain the technical novelty of the paper. They can also carry out more exhaustive empirical evaluations and provide more intuition into the key concepts of the paper to improve the quality of the paper as suggested in the weakness section.

**Limitations:**

The paper mentions some limitations related perspectives in the conclusion section.

---

> ### Author Rebuttal · Authors · 2023-08-09
>
> We appreciate your detailed feedback on our paper. In response to the raised concerns about novelty, experimental evaluation, and clarity, we provide detailed explanations below.
>
> **W1.**
> We wish to highlight our main novelty, which is identifying the research gap arising from the asymmetry between positive and negative samples in healthcare analytics - a subject that prior work has not adequately explored. It is non-trivial to pinpoint this gap and address it in an effective manner. As front-line clinicians and medical researchers, we propose a systematic and innovative solution to explore negative samples for cohort discovery, including Negative Sample Shapley Field construction, manifold learning with structure preservation and isotropy constraint, and cohort discovery among high data Shapley value negative samples. Our proposal has undergone both an empirical evaluation and an in-depth clinical validation that demonstrates promising results in unveiling meaningful insights for cohort discovery.
>
> **W2.**
> We have strengthened our experimental evaluation with three supplemented comparison studies:
> * We have introduced contrastive PCA as a baseline for cohort discovery, and the results are presented in Figure B. Detailed explanations can be found in our response to Weakness 1 of Reviewer vE7H's reviews.
> * We have benchmarked our proposal against three positive-unlabelled learning methods, Classic Elkanoto, Weighted Elkanoto, and Bagging-based PU-learning. The comparison results are illustrated in Figure C, and further details are provided in our response to Weakness 1 of Reviewer g4r8's reviews.
> * We have conducted an ablation study to validate the influence of the isotropy constraint, and the results are displayed in Figure A, with a comprehensive analysis in our response to Weakness 2 of Reviewer g4r8's reviews.
>
> The experimental results of these three studies collectively affirm the efficacy of our proposal from different perspectives.
>
> It is essential to highlight that our focus as front-line clinicians (specifically nephrologists) and medical researchers lies in investigating cohort discovery among negative samples using our hospital's EMR data of our patients, with a specific emphasis on hospital-acquired AKI, which we strive to handle in practice. As a result, we are able to provide accurate interpretation and conduct in-depth clinical validation, leading to meaningful medical insights and benefits for patients and clinicians in the long term. While evaluating our proposal on other datasets is feasible, it remains highly challenging to derive medically meaningful results for cohort discovery, since specialized medical expertise and understanding of the context is indispensable.
>
> Our motivation stems from addressing our real-world medical problems in clinical practice, which adds a unique perspective to the design of our methodology for cohort discovery. We acknowledge the potential applicability and generalizability of our identified problem and proposed method to other domains. Although beyond the current scope of our research, we are open to considering evaluations in other domains for future work.
>
> **W3.**
> In cooperative game theory, the Shapley value is a widely recognized solution for transferable utility cooperative games. These games consist of a player set and a characteristic function that assigns values to coalitions (subsets of players). In such games, the Shapley value offers a rigorous and intuitive way to fairly distribute the collective value of the team across individuals [38].
>
> Building upon the Shapley value, the data Shapley value presents an established approach for addressing the cooperative game of data valuation [12]. In this context, the player set represents data samples, and the characteristic function is defined as the payoff of the model, typically measured by goodness-of-ﬁt metrics such as accuracy or AUC. This is the underlying rationale of data Shapley values for data valuation.
>
> In our work, we frame our problem as a negative sample valuation game to ensure an equitable distribution of the collective performance achieved by the prediction model to each participating negative sample in the training data. To achieve this objective, we defined data Shapley values for negative samples in Proposition 1. Subsequently, we creatively utilize the distribution of our defined data Shapley values for negative samples as a means of achieving cohort discovery, taking into account the notion that diverse cohorts should exhibit varying distributions. Through experimental evaluation, we showcase the efficacy of our proposal in segregating high data Shapley value samples into discrete, medically pertinent cohorts, as illustrated in Figure 6. This successful stratification offers valuable medical insights of clinical relevance.
>
> Further, the distribution of negative samples with similar data Shapley values is nonuniform due to the complexity of real-world EMR data, such as varying scales of medical features (e.g., log scale) and the diverse influence of feature value ranges on the data Shapley value. In addition, this nonuniform distribution was also identified in our preliminary experiments, where we found that without the isotropy constraint to ensure that data Shapley value changes are uniform across orientations, we could not discover any meaningful cohort discovery results in subsequent spatial clustering (as illustrated in Figure A(b)). To address this issue, we propose to integrate the isotropy constraint in the representation learning process for cohort discovery. Please refer to our response to Weakness 2 of Reviewer g4r8's reviews for more details on the impact of the isotropy constraint.
>
> **Questions.**
> Please refer to our responses to Weaknesses 1, 2, and 3 highlighted in your reviews for further clarification. We hope that these address the concerns effectively and strengthen the overall quality and contributions of our paper.

---

> > ### Comment · Area_Chair_YcGK · 2023-08-18
> > **Response to author rebuttal**
> >
> > Dear reviewer,
> >
> > The author rebuttal appears to have presented several targeted responses to your questions.
> >
> > Are your questions appropriately addressed?
> > If they are, would you consider re-assessing your score in light of them.
> > If not, please do provide additional context and feedback to the author.
> >
> > In either case, please provide an acknowledgement of the effort the authors put in, why your questions have (or have not) been addressed and what your assessment of the work is in light of this evidence with a view to reach consensus with the other reviewers on this work.
> >
> > -AC

---

> > ### Comment · Reviewer_vvb4 · 2023-08-20
> > **Thank you for your response!**
> >
> > Thank you for getting back to me regarding my questions! I am not convinced by the arguments regarding the novelty of the paper.  Cohort discovery is a long-standing problem in health analytics, and the asymmetry between positive and negative examples is also a well-aware problem in various applications of machine learning. In addition, I am also not sure if the technical novelty of the paper is sufficient.
> >
> > Thank you also for providing additional experiments and clarification. It seems that incorporating these updates will require a substantial revision of this paper. As such, I am not sure if the paper is ready for publication at this time.

---

> > > ### Author Response · Authors · 2023-08-21
> > > **Thanks for your reply**
> > >
> > > Thanks for your reply. We understand your concern regarding the novelty of our study, and therefore, we intend to provide further elaboration on this aspect.
> > >
> > > Cohort analysis and cohort discovery are undoubtedly compelling research directions. Progress has been made in enhancing the quality of embeddings to improve task performance and enhancing model interpretability. However, there is considerable potential for further exploration in this research area.
> > >
> > > **In contrast to mainstream medical cohort-related research, our study adopts a novel perspective.** Specifically, we focus on cases/samples that have undergone medical examinations but have not received a clinical diagnosis for a specific disease. These samples are considered negative or unlabeled. This perspective highlights the distinct nature of medical diagnosis compared to traditional classification problems, e.g., medical diagnoses are grounded in designated medical criteria that rely on medical theories and experience, evolving dynamically with deeper disease understanding or societal changes. Consequently, negative samples in the medical domain hold significant value for research. We emphasize the data Shapley values of negative samples, which are particularly important due to their potential to reveal future positives, pathological correlations, or conditions that bear similarities. This reciprocal relationship between negative and positive samples contributes to defining positive samples in theoretical medical research. **Importantly, prior work has not explored this valuable perspective with significant medical potential.**
> > >
> > > This paper pioneers the exploration of medical classification problems from this unique perspective. While our primary focus is not on rectifying the asymmetry between positive and negative examples for enhanced performance, our techniques do have relevant implications. Responding to the suggestion of Reviewer g4r8, we have included comparisons with state-of-the-art positive-unlabeled learning baselines. Experimental results (Figure C) establish the superiority of our approach in identifying negative samples in real-world medical data. **This validation underscores the validity of our constructed Negative Sample Shapley Field, forming a robust foundation for subsequent cohort discovery.**
> > >
> > > Regarding Shapley values or data Shapley values, existing studies predominantly measure the value of individual data samples (primarily for federated learning contributions) or apply them at a finer level (e.g., in feature interpretation). **What distinguishes our paper is its innovative extension of this concept to Shapley-based exploration of interrelationships between samples. This extension moves beyond traditional feature-based similarity methods, asserting that valuable cohorts should exhibit similar distributions with high data Shapley values.**
> > >
> > > The novelty of our constructed Negative Sample Shapley Field for clustering analysis lies not only in accounting for feature similarity but also in imposing data Shapley value constraints. Given the challenges of density-based spatial clustering, we propose manifold learning while concurrently integrating the isotropy constraint to address the issue effectively. **Supplementary experiments have demonstrated the efficacy of the isotropy constraint through an ablation study (Figure A).**
> > >
> > >
> > >
> > >
> > > Furthermore, as the experimental results illustrate, our proposal outperforms traditional algorithms rooted in sample features only (cPCA) in clustering analysis, as evidenced by t-SNE (Figure B). This success reaffirms the effectiveness of our innovative approach - leveraging data Shapley values for cohort discovery. Additional experiments highlight that after manifold learning on the constructed Negative Sample Shapley Field, our proposal detects distinctive clustering outcomes, resulting in **robust cohort discovery outcomes resistant to spatial clustering algorithm parameter variations (Figure A (a)).**
> > >
> > > Clinically validated by medical professionals, our derived cohort discovery results validate the correctness of the outcomes and the medical utility of our proposal. Concurrently, as shown in Figure 6, our discovered cohorts adeptly decompose the overall distribution into distinct, medically relevant groups, represented in distinct Gaussian components. **This validates the efficacy of the hot zone search in our approach and, crucially, demonstrates that our data Shapley value-based strategy serves as a sturdy foundation for cohort discovery and sample clustering analysis.**
> > >
> > > While limitations during the author response phase preclude us from presenting a revised manuscript, we are confident that our enhancements, as suggested by reviewers during this phase, have been thoroughly discussed above.

---

### Official Review · Reviewer_g4r8 · 2023-07-09

**Soundness:** 3 good
**Presentation:** 3 good
**Contribution:** 3 good
**Rating:** 7
**Confidence:** 3

**Summary:**

This paper describes a method to understand the set of unlabelled / negative samples in a healthcare data-set. In this setting, one typically has a set of patients with a particular label, such as indicidence of a particular disease, and a large set of unlabelled samples. Training a classifier involves selecting some subset of the unlabelled samples as the set of negative samples for training. This set is often quite heterogeneous, so methods that enable better understanding of the structure in the data and selection of negative samples can be informative.

The key contributions are as follows:
1. Definition of the Negative Shapley Value Field, which associates the Shapley Value for the the prediction task of interest with each negative sample
2. Illustration of a representation learning method which discovers a low-dimensional representation for the negative samples in which samples with similar Shapley Values are close to one another
3. A method for cohort discovery based on clustering in the low-dimensional space and interpretive analysis to demonstrate the clinical coherence and relevance of the discovered cohorts
4. Demonstration of improved predictive performance when selecting samples based on the negative Shapley value
5. Demonstration that predictive performance is maintained when the low-dimensional representations are used

**Strengths:**

The problem of understanding and selective negative samples for cohorts in healthcare data is an important one and methods that can be used by practitioners in this domain will be valuable.

The authors provide a demonstration of the utility in improved classifier performance when limiting to the set of negative samples with Shapley value > 0. This is suggested to be due to those samples which have negative Shapley value corresponding to patients who are likely to present with AKI in the future, but do not yet have this information in their medical record. This is a nice result and tackles a common problem in biomedical data science.

Mapping the unlabelled samples into a low-dimensional space where samples with similar representations are expected to have similar Shapley values is shown to enable the discovery of distinct and interpretable cohorts of samples with similar features and similar Shapley values. This result could provide a useful tool for practitioners to select or filter the set of negative samples when building classifiers on EHR data

**Weaknesses:**

*** These weaknesses have been addressed in the author response ***

The "Effectiveness of the Negative Sample Shapley Field", described in lines 316-322, is illustrated by showing that filtering the set of unlabelled samples to exclude those which had a negative Shapley value improves the performance of the trained classifier on held-out data reminded me of co-training [1] or positive-unlabelled learning [2]. It would have been interesting to see this approach benchmarked against other methods for developing classifiers based on positive and unlabelled data, where we expect a number of the unlabelled samples to be positive rather than negative samples

The "Effectiveness of Cohort Discovery", described on lines 332-345 is a nice result but it is not clear from the experiments to what degree the isotropy constraint enabled this. This could be demonstrated by an experiment in which the same SDAE model is applied to the data  without the isotropy constraint.

[1] Blum, A., Mitchell, T. Combining labeled and unlabeled data with co-training. COLT: Proceedings of the Workshop on Computational Learning Theory, Morgan Kaufmann, 1998, p. 92-100.
[2] Bekker, J., Davis, J. Learning from positive and unlabeled data: a survey. Mach Learn 109, 719–760 (2020). https://doi.org/10.1007/s10994-020-05877-5

**Questions:**

Lines 316-322, explore the effectiveness of the constructed Negative Sample Shapley Field. Is it fair to say that this experiment supports the effectiveness of using Shapley Values of a set of unlabelled samples to eliminate samples with false negative labels and that this is independent of the concept of the Shapley Value Field, which considers the association between input features and Shapley value for each sample in the set of unlabelled samples.

**Limitations:**

*** These concerns have been addressed in the author response ***

This is an interesting paper with some results which could be useful in biomedical data science, but which would be made more convincing by more thorough benchmarking and comparison with other approaches to achieve each of their key results.

---

> ### Author Rebuttal · Authors · 2023-08-09
>
> We thank the reviewer for the positive feedback and valuable advice, particularly the suggestion to benchmark our proposal against positive-unlabelled learning methods, which has greatly contributed to enhancing our paper. As for the several weaknesses and questions you mentioned, our responses are listed as follows.
>
> **W1.**
> We have selected three influential positive-unlabelled learning methods from the recommended references, namely Classic Elkanoto [c], Weighted Elkanoto [c], and Bagging-based PU-learning [d]. Following the implementation in [e], we train the three baseline methods that consider negative samples as unlabeled data and evaluate their performance on the testing data.
>
> The experimental results of these baseline methods in terms of AUC (mean $\pm$ std) from five repeats are presented in Figure C. Among the three newly benchmarked baselines, Bagging-based PU-learning outperforms the other two methods and also surpasses the performance of the "All $d_i^-$" setting, where all negative/unlabeled samples are included in the training. This validates the effectiveness of Bagging-based PU-learning in positive-unlabelled learning, achieved through its bootstrap aggregating techniques. On the other hand, both Classic Elkanoto and Weighted Elkanoto fail to achieve satisfactory performance. They merely marginally outperform the "All $d_i^-$" setting when adopting LR and AdaBoost. This observation suggests that the "selected completely at random" assumption adopted by these two baselines might not hold in our hospital-acquired AKI prediction using real-world EMR data. In contrast to all these baselines, the "$d_i^-$ with $s_i>0$" setting of our proposal, which filters out the negative samples with negative data Shapley values, consistently achieves significantly higher AUC values across different classifiers. This confirms the superiority of our proposed Negative Sample Shapley Field in effectively handling negative/unlabelled samples over the benchmarked baselines.
>
> The experimental results in comparison with positive-unlabelled learning methods above serve to validate the effectiveness of our proposal, particularly the constructed Negative Sample Shapley Field. We further highlight that different from these positive-unlabelled learning methods, our main objective in this work lies in achieving effective cohort discovery among negative samples rather than solely aiming to boost analytic performance.
>
> [c] Elkan, Charles, and Keith Noto. "Learning classifiers from only positive and unlabeled data." Proceedings of the 14th ACM SIGKDD international conference on Knowledge discovery and data mining. 2008.
>
> [d] Mordelet, Fantine, and J-P. Vert. "A bagging SVM to learn from positive and unlabeled examples." Pattern Recognition Letters 37 (2014): 201-209.
>
> [e] https://github.com/pulearn/pulearn
>
> **W2.**
> We have supplemented an ablation study to investigate the impact of removing the isotropy constraint from our proposed method. First, we demonstrate the cohort discovery results of our proposal in Figure A(a), varying the $P_{min}$ settings. We observe that the results remain relatively stable across different $P_{min}$ settings, leading us to set $P_{min}=100$ in Figure 3. For comparison, we present the cohort discovery results of the weakened version of our proposal, i.e., without the isotropy constraint, in Figure A(b). It is clearly shown that the absence of the isotropy constraint leads to ineffective cohort discovery through DBSCAN, irrespective of the $P_{min}$ settings. Hence, this study affirms the critical role of the isotropy constraint in enabling meaningful and interpretable cohort discovery, ensuring the identification of relevant and valuable patterns within the data.
>
> We would like to further share our prior experience during the development of our cohort discovery method. Initially, we did not include the isotropy constraint in our proposal. After constructing the Negative Sample Shapley Field, we conducted manifold learning without the isotropy constraint and used DBSCAN for cohort discovery. However, we encountered challenges in deriving meaningful cohorts from DBSCAN, and the corresponding t-SNE plots were incomprehensible to medical experts (similar to Figure A(b)). Upon investigating this issue, we identified that DBSCAN being a spatial clustering method, requires proper handling of spatial information in the Negative Sample Shapley Field to achieve effective cohort discovery. To address this, we introduced the isotropy constraint to ensure uniform data Shapley value changes across orientations, rendering our proposal more amenable to subsequent spatial clustering. As a result, we successfully avoided the mis-discovery, as exemplified in Figure 1(b) and consequently, contributed to unveiling medically meaningful cohorts.
>
> **Questions.**
> Given the exponential complexity of computing the data Shapley values for negative samples as defined in Eq.1, we propose to employ Monte Carlo permutation sampling for approximation in Eq.2. The purpose of this experiment is to validate the functionality of our approximated data Shapley values for negative samples, ensuring they align with our expectations. The positive outcome of this experiment underpins the construction of the Negative Sample Shapley Field, which plays a pivotal role in our cohort discovery methodology.
>
> **Limitations.**
> We have integrated three positive-unlabelled learning methods, namely Classic Elkanoto, Weighted Elkanoto, and Bagging-based PU-learning, for comparison with our proposal. Additionally, we have conducted an ablation study to assess the influence of the isotropy constraint in our proposed cohort discovery method. We kindly direct you to our response addressing Weaknesses 1 and 2 outlined in your reviews.
>
> Furthermore, we have compared our proposal with contrastive PCA in cohort discovery. Detailed information on this comparison can be found in our response to Weakness 1 of Reviewer vE7H's reviews.

---

> > ### Comment · Area_Chair_YcGK · 2023-08-18
> > **Response to author rebuttal**
> >
> > Dear reviewer,
> >
> > The author rebuttal appears to have presented several targeted responses to your questions.
> >
> > Are your questions appropriately addressed?
> > If they are, would you consider re-assessing your score in light of them.
> > If not, please do provide additional context and feedback to the author.
> >
> > In either case, please provide an acknowledgement of the effort the authors put in, why your questions have (or have not) been addressed and what your assessment of the work is in light of this evidence with a view to reach consensus with the other reviewers on this work.
> >
> > -AC

---

> > ### Comment · Reviewer_g4r8 · 2023-08-18
> > **Response to author rebuttal**
> >
> > I thank the authors for their response. It's great to see that their method, which leverages some additional assumptions on the unlabelled samples, outperforms the best-performing PU learning approach. I have increased my score

---

> > > ### Author Response · Authors · 2023-08-18
> > > **Thanks for your reply**
> > >
> > > Thank you for acknowledging our newly supplemented experiments and for the significant increase in rating. We highly appreciate your constructive feedback.

---

### Official Review · Reviewer_vE7H · 2023-07-10

**Soundness:** 2 fair
**Presentation:** 2 fair
**Contribution:** 3 good
**Rating:** 5
**Confidence:** 5

**Summary:**

In healthcare analytics, cohort constructions is one of the key steps that drives the analysis. For most problems, where the outcome of interest is a disease, the problem has asymmetrical formalism - while patients with disease are defined using string criterion and are homogenous w.r.t problem the negative set can be diverse and can have important information that is under-analyzed. The authors present a Shapley value driven approach to analyze the negative set in terms of their contribution to the predictive power of the models. Furthermore, these mappings are transformed and clustered to identify potentially clinical important patients. They have presented results and commentary from clinicians on identified clusters.

**Strengths:**

- The authors raise an interesting hypothesis about under-analysis of the negative samples that can drive the community to develop standard methods to handle such problems
- It is commendable that the authors validated their findings and presented commentaries from clinicians on the identified patterns. Such efforts are increasingly important to ascertain the clinical validity of proposed AI methods
- Overall, the intuition behind the method is novel and somewhat defensible. The authors have also made an effort to formalize many aspects of their approach
- The authors have also made an effort to validate the components of the method individually (see more on this below)

**Weaknesses:**

- The primary weakness of the paper is a lack of comparison against baseline methods that necessitates the complexity of the proposed methods. There is also a lack of studying the correctness of the proposed cohort discovery method. It may be beneficial for the authors to support their claim on a synthetic datasets and/or provide comparisons of discovered cohorts using other standard methods such as contrastive PCA.
- Continuing from the above, the computational complexity of the proposed approach hasn't been acknowledged in a satisfactory manner. While Monte Carlo methods have been proposed to calculate the values, the true complexity in evaluating over the entire negative set and the subsequent calculations imposed the isotropy constraints hasn't been analyzed clearly.

Edit: The authors have responded by providing additional baseline comparisons that alleviates some of the concerns. I have updated my review to reflect the same

**Questions:**

See above

---

> ### Author Rebuttal · Authors · 2023-08-09
>
> We would like to express our gratitude for your constructive comments regarding our paper, especially the suggestion to compare our method with contrastive PCA. We have incorporated the recommended experiment and provided a comprehensive complexity analysis. Please find our detailed responses addressing each of your comments below.
>
> **W1.**
> We would like to clarify that synthetic datasets are typically generated based on known assumptions, whereas our primary objective is to uncover novel insights from our hospital's EMR data to provide valuable information to AKI (acute kidney injury) for the benefit of our patients and clinicians. To evaluate the correctness of our proposed cohort discovery method, we approach the assessment from two perspectives. Firstly, we illustrate the data Shapley value histogram of the cohorts identified using our method, as depicted in Figure 6. This validation demonstrates our method's capability to decompose high data Shapley value samples into distinct, medically relevant cohorts. Secondly, we perform an in-depth analysis of the discovered cohorts from the medical perspective.
>
> We appreciate the reviewer's suggestion of contrastive PCA (cPCA), the key idea of which is to utilize a background dataset to cancel out the common patterns in the target dataset and therefore, reveal the unique patterns in the target dataset [a, b]. We agree that cPCA shares a similar spirit with our proposal. Therefore, we have conducted a comparison with cPCA for cohort discovery. Specifically, on account of our target to discover cohorts among negative samples, we treat the negative samples as the target dataset and the positive samples as the background dataset. We then retain 64 contrastive principal components in the projected data, maintaining the same dimension as the output of our SDAE. Subsequently, we apply DBSCAN to the projected data derived by cPCA for cohort discovery. The experimental results of cPCA for four $\alpha$ values ($0$, $1.06$, $5.54$, and $74.44$) automatically selected by cPCA are presented in Figure B, and the cohort discovery results of our proposal are illustrated in Figure A(a). The comparison reveals that our proposal outperforms cPCA (across different $\alpha$ values) in identifying medically meaningful cohorts for various $P_{min}$ values (the minimum number of points in a neighborhood to define a core point in DBSCAN).
>
> In addition, we have benchmarked our proposal against three positive-unlabelled learning methods: Classic Elkanoto, Weighted Elkanoto, and Bagging-based PU-learning. For a detailed account of this benchmarking study, we kindly refer you to our response addressing Weakness 1 of Reviewer g4r8's reviews.
>
> [a] Abid, Abubakar, et al. "Contrastive principal component analysis." arXiv preprint arXiv:1709.06716 (2017).
>
> [b] Abid, Abubakar, et al. "Exploring patterns enriched in a dataset with contrastive principal component analysis." Nature communications 9.1 (2018): 2134.
>
> **W2.**
> We analyze the complexity of our proposed cohort discovery method in a step-wise manner as outlined below. Here, $f$ represents the feature dimension, $N$ is the total number of samples, and $N^{-}$ is the number of negative samples. Since $N$ and $N^{-}$ are on the same order, we use $N$ consistently in the following complexity analysis for simplicity.
>
> **Step 1. Negative Sample Shapley Field Construction.** We employ Monte Carlo permutation sampling to calculate the data Shapley values for negative samples. Each Monte Carlo permutation involves using the logistic regression model to compute the data Shapley value, with AUC as the evaluation metric. The complexity of logistic regression is $O(Nf)$, while the complexity of AUC calculation is $O(N\log N)$. As suggested by [12], the convergence of Monte Carlo permutation sampling is generally reached with number of samples on the order $N$, and in our experiments, we run over $5N$ permutations. Thus, the complexity of Step 1 is $O(N^{2}(f+\log N))$.
>
> **Step 2. Manifold Learning with Structure Preservation and Isotropy Constraint.** Our SDAE consists of $K$ DAEs, where the input dimension of the $k$-th DAE's encoder is $n_k$ and the output dimension is $m_k$. The overall complexity of SDAE is $O(N \sum_{k=0}^{K-1}(n_k m_k))$, which could be simplified to $O(N)$, considering that $K$, $n_k$ and $m_k$ are constants.
> Next, for the isotropy constraint, we calculate the distance between each pair of samples within each batch, resulting in ${|\mathcal{B}|  \choose  2} =|\mathcal{B}| (|\mathcal{B}| -1)/2$  distance calculations. Since the complexity of computing the distance between each pair of samples is $O(f)$, the complexity of imposing the isotropy constraint for each batch is $O((|\mathcal{B}| (|\mathcal{B}| -1)/2) f) = O(|\mathcal{B}|^2f)$. With a total of $N/|\mathcal{B}|$ batches, the overall complexity of imposing the isotropy constraint is $O(N|\mathcal{B}|f)$. Combining the computation in both the SDAE and the isotropy constraint, the complexity of Step 2 is $O(N|\mathcal{B}|f)$.
>
> **Step 3. Cohort Discovery Among High Data Shapley Value Negative Samples.** In this step, we mainly conduct DBSCAN on all the negative samples. The average complexity of this process is $O(N \log N)$ with the use of an accelerating index structure, while the worst-case complexity is $O(N^{2})$.
>
> In summary, our proposed cohort discovery method with all three aforementioned steps exhibits an overall complexity of $O(N^{2}\log N)$, with the major computational overhead occurring in Step 1. During this step, we calculate the data Shapley values for negative samples using Monte Carlo permutation sampling in $O(N)$ time, and subsequently compute the AUC metric of logistic regression for each permutation in $O(N\log N)$ time.

---

> > ### Comment · Reviewer_vE7H · 2023-08-17
> > **Thanks for the response**
> >
> > Thanks for the response, especially for taking the time to provide detailed additional baseline experiments. Overall, the additional results provides better proof for the claims of the paper. I have updated the review to reflect the same.
> >
> > For future improvements, the authors may want to understand the exact properties of shapley values that makes the approach suitable for the use case. Essentially, while this is currently a nice application, in the current state it is difficult to evaluate how well and on which class of the problem the approach will generalize to

---

> > > ### Author Response · Authors · 2023-08-18
> > > **Thanks for your reply**
> > >
> > > Thanks for your feedback. We highly appreciate that you take our new experimental results into consideration and reflect these in your updated review.
> > >
> > > Motivated by the asymmetry between positive and negative samples, we propose to leverage data Shapley values to quantify the impact of negative samples, in our application of facilitating cohort discovery in order to reveal novel medical insights in the realm of healthcare analytics.
> > >
> > > We value your constructive recommendations for further improvement and invite deeper discourse. Recognizing your reservations about the current experimental evaluation, we are earnestly seeking avenues to broaden the application scenarios, ensuring medical relevance. Your guidance on supplementary evaluations that would elevate the quality of our current evaluation would be greatly esteemed.

---

### Author Rebuttal · Authors · 2023-08-09

We would like to extend our profound gratitude to all the reviewers for their insightful comments and constructive suggestions, which have played a pivotal role in elevating the quality of our paper. In this global response, we wish to first emphasize the primary novelty and contributions of our work. Subsequently, we provide a concise overview of the improvements made to the experimental evaluation. Furthermore, we address each reviewer's comments individually and in detail in the separate responses below.

**Novelty and Contributions.**

We pioneer the recognition of a previously unexplored research void arising from the asymmetry between positive and negative samples in healthcare analytics. This oversight is crucial, especially when negative samples can offer as much, if not more, insight as their positive counterparts. To address this, we creatively introduce the concept of data Shapley values for negative samples. We posit that the distribution of these values can be instrumental in guiding subsequent cohort discovery, premised on the notion that distinct cohorts will manifest different distributions.

Driven by this realization, we devise a comprehensive and novel approach for identifying cohorts within negative samples. This involves the establishment of the Negative Sample Shapley Field, the integration of manifold learning ensuring structure preservation and isotropy constraint, and the segregation of high data Shapley value negative samples into discernible cohorts.

Our dual role as clinicians and medical researchers places us in a unique position to validate our methodology. Utilizing patients' EMR data from our hospital, we undertake a rigorous evaluation. The experimental results not only attest to the effectiveness of the individual components of our proposal but also underscore its proficiency in decomposing high data Shapley value samples into distinct, medically-significant cohorts, thereby unveiling clinical insights of paramount importance.

**Enhancement of Experimental Evaluation.**

In response to the valuable feedback from the reviewers, we have augmented our experimental evaluation by integrating three additional comparative studies:

(i) A direct comparison of our proposal's cohort discovery results against those of contrastive PCA is depicted in Figure B.
This comparative visualization clearly underscores that our proposal consistently identifies cohorts that are more information-rich than those determined by contrastive PCA across various $\alpha$ settings, demonstrating the superiority of our proposal in cohort discovery.

(ii) We have adopted three positive-unlabelled learning methods, Classic Elkanoto, Weighted Elkanoto, and Bagging-based PU-learning, as baselines for comparison. The experimental results are presented in Figure C, which shows when we employ the "$d_i^-$ with $s_i>0$" setting (which excludes negative samples with negative data Shapley values),  our proposal persistently surpasses all three baselines in AUC by a large margin. This affirms the efficacy of our proposal in handling negative/unlabelled samples.

(iii) To evaluate the impact of our introduced isotropy constraint in manifold learning, we have conducted an ablation study with the results displayed in Figure A. The absence of this isotropy constraint leads our proposed cohort discovery method to fall short in discerning any meaningful cohorts across diverse $P_{min}$ configurations. This underscores the critical role of the isotropy constraint in ensuring effective cohort discovery.

We believe that the aforementioned revisions have substantively enhanced the quality and rigor of our paper, which we hope will meet the high standards of the reviewers and contribute meaningfully to the field of machine learning for healthcare. If there are any additional questions or comments, we stand ready and eager to engage in further discussions.

---

### Decision · Program_Chairs · 2023-09-21

**Decision:**

Reject

**Comment:**

This work describes a set of steps to identify the right unlabelled examples (as negative samples) in a healthcare data-set for a predictive task. The paper calculates the data Shapley value of the negative samples in the dataset. Then a stacked DAE is trained to capture representations that cluster the ShapValues in the negative samples. The paper studies the improvement in predictive performance when selecting samples based on negative Shap values and the low-dimensional representations. The initial stage of reviews highlighted the lack of relevant baselines performed for the method and the clarity of writing. During the rebuttal, the paper was augmented with additional experiments (specifically, a comparison with cPCA, PU learning and an analysis of the complexity of the work). At the end of the review period, there was a lack of consensus among the reviewers. One reviewer highlighted the computational cost associated with the methods and the resulting infeasibility of applying this class of methods to large scale models. Overall, I do think the problem considered herein is solid and the method proposed is simple but there still remains work to be done to bring this paper up to publication. In particular, I think an appropriate acknowledgement of the costs of the method are in order and a study of the tradeoffs in performance vs compute cost against the baselines added during the rebuttal would go a long way towards improving the manuscript.